# Unsupervised learning of object frames by dense equivariant image labelling

**James Thewlis**[1]          **Hakan Bilen**[2]          **Andrea Vedaldi**[1]

[1] Visual Geometry Group
University of Oxford
`{jdt,vedaldi}@robots.ox.ac.uk`

[2] School of Informatics
University of Edinburgh
`hbilen@ed.ac.uk`

## Abstract

One of the key challenges of visual perception is to extract abstract models of 3D objects and object categories from visual measurements, which are affected by complex nuisance factors such as viewpoint, occlusion, motion, and deformations. Starting from the recent idea of viewpoint factorization, we propose a new approach that, given a large number of images of an object and no other supervision, can extract a dense object-centric coordinate frame. This coordinate frame is invariant to deformations of the images and comes with a dense equivariant labelling neural network that can map image pixels to their corresponding object coordinates. We demonstrate the applicability of this method to simple articulated objects and deformable objects such as human faces, learning embeddings from random synthetic transformations or optical flow correspondences, all without any manual supervision.

## 1 Introduction

Humans can easily construct mental models of complex 3D objects and object categories from visual observations. This is remarkable because the dependency between an object's appearance and its structure is tangled in a complex manner with extrinsic nuisance factors such as viewpoint, illumination, and articulation. Therefore, learning the intrinsic structure of an object from images requires removing these unwanted factors of variation from the data.

The recent work of [39] has proposed an unsupervised approach to do so, based on on the concept of *viewpoint factorization*. The idea is to learn a deep Convolutional Neural Network (CNN) that can, given an image of the object, detect a discrete set of object landmarks. Differently from traditional approaches to landmark detection, however, landmarks are neither defined nor supervised manually. Instead, the detectors are learned using only the requirement that the detected points must be equivariant (consistent) with deformations of the input images. The authors of [39] show that this constraint is sufficient to learn landmarks that are "intrinsic" to the objects and hence capture their structure; remarkably, due to the generalization ability of CNNs, the landmark points are detected consistently not only across deformations of a given object instance, which are observed during training, but also across different instances. This behaviour emerges automatically from training on thousands of single-instance correspondences.

In this paper, we take this idea further, moving beyond a sparse set of landmarks to a dense model of the object structure (section 3). Our method relates each point on an object to a point in a low dimensional vector space in a way that is consistent across variation in motion and in instance identity. This gives rise to an object-centric coordinate system, which allows points on the surface of an object to be indexed semantically (figure 1). As an illustrative example, take the object category of a face and the vector space $\mathbb{R}^3$. Our goal is to semantically map out the object such that any point on a face, such as the left eye, lives at a canonical position in this "label space". We train a CNN to learn the function that projects any face image into this space, essentially "coloring" each pixel with its

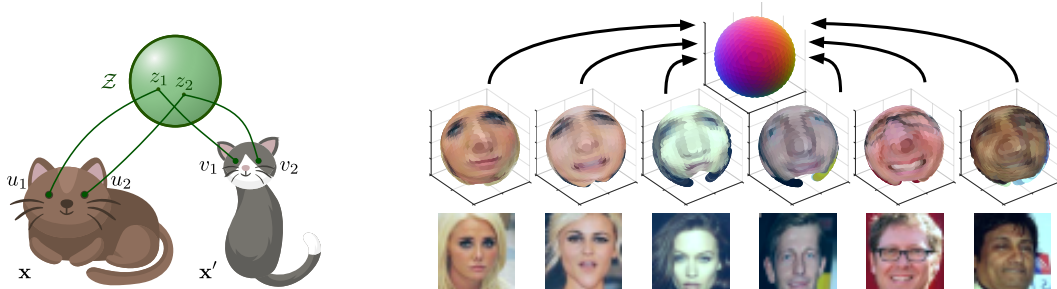

Figure 1: **Dense equivariant image labelling.** *Left:* Given an image **x** of an object or object category and no other supervision, our goal is to find a common latent space $\mathcal{Z}$, homeomorphic to a sphere, which attaches a semantically-consistent coordinate frame to the object points. This is done by learning a dense labelling function that maps image pixels to their corresponding coordinate in the $\mathcal{Z}$ space. This mapping function is equivariant (compatible) with image warps or object instance variations. *Right:* An equivariant dense mapping learned in an unsupervised manner from a large dataset of faces. (Results of SIMPLE network, $\mathcal{L}_{dist}, \gamma = 0.5$)

corresponding label. As a result of our learning formulation, the label space has the property of being locally smooth: points nearby in the image are nearby in the label space. In an ideal case, we could imagine the surface of an object to be mapped to a sphere.

In order to achieve these results, we contribute several technical innovations (section 3.2). First, we show that, in order to learn a non-trivial object coordinate frame, the concept of equivariance must be complemented with the one of *distinctiveness* of the embedding. Then, we propose a CNN implementation of this concept that can explicitly express uncertainty in the labelling of the object points. The formulation is used in combination with a probabilistic loss, which is augmented with a robust geometric distance to encourage better alignment of the object features.

We show that this framework can be used to learn meaningful object coordinate frames in a purely unsupervised manner, by analyzing thousands of deformations of visual objects. While [39] proposed to use Thin Plate Spline image warps for training, here we also consider simple synthetic articulated objects having frames related by known optical flow (section 4).

We conclude the paper with a summary of our finding (section 5).

## 2   Related Work

**Learning the structure of visual objects.** Modeling the structure of visual objects is a widely-studied (*e.g.* [6, 7, 11, 41, 12]) computer vision problem with important applications such as facial landmark detection and human body pose estimation. Much of this work is supervised and aimed at learning detectors of objects or their parts, often using deep learning. A few approaches such as spatial transformer networks [20] can learn geometric transformations without explicit geometric supervision, but do not build explicit geometric models of visual objects.

More related to our work, WarpNet [21] and geometric matching networks [35] learn a neural network that predicts Thin Plate Spline [3] transformations between pairs of images of an object, including synthetic warps. Deep Deformation Network [44] improves WarpNet by using a Point Transformer Network to refine the computed landmarks, but it requires manual supervision. None of these works look at the problem of learning an invariant geometric embedding for the object.

Our work builds on the idea of *viewpoint factorization* (section 3.1), recently introduced in [39, 32]. However, we extend [39] in several significant ways. First, we construct a *dense* rather than discrete embedding, where all pixels of an object are mapped to an invariant object-centric coordinate instead of just a small set of selected landmarks. Second, we show that the equivariance constraint proposed in [39] is not quite enough to learn such an embedding; it must be complemented with the concept of a *distinctive* embedding (section 3.1). Third, we introduce a new neural network architecture and corresponding training objective that allow such an embedding to be learned in practice (section 3.2).

**Optical/semantic flow.** A common technique to find correspondences between temporally related video frames is optical flow [18]. The state-of-the-art methods [14, 40, 19] typically employ convolu-

tional neural networks to learn pairwise dense correspondences between the same object instances at subsequent frames. The SIFT Flow method [25] extends the between-instance correspondences to cross-instance mappings by matching SIFT features [27] between semantically similar object instances. Learned-Miller [24] extends the pairwise correspondences to multiple images by posing a problem of alignment among the images of a set. Collection Flow [22] and Mobahi *et al.* [29] project objects onto a low-rank space that allow for joint alignment. FlowWeb [52], and Zhou *et al.* [51] construct fully connected graphs to maximise cycle consistency between each image pair and synthethic data as an intermediary by training a CNN. In our experiments (section 4) flow is known from synthetic warps or motion, but our work could build on any unsupervised optical flow method.

**Unsupervised learning.** Classical unsupervised learning methods such as autoencoders [4, 2, 17] and denoising autoencoders aim to learn useful feature representations from an input by simply reconstructing it after a bottleneck. Generative adversarial networks [16] target producing samples of realistic images by training generative models. These models when trained joint with image encoders are also shown to learn good feature representations [9, 10]. More recently several studies have emerged that train neural networks by learning auxiliary or pseudo tasks. These methods exploit typically some existing information in input as "self-supervision" without any manual labeling by removing or perturbing some information from an input and requiring a network to reconstruct it. For instance, Doersch *et al.* [8], and Noroozi and Favaro [31] train a network to predict the relative locations of shuffled image patches. Other self-supervised tasks include colorizing images [46], inpainting [34], ranking frames of a video in temporally correct order [28, 13]. More related to our approach, Agrawal *et al.* [1] use egomotion as supervisory signal to learn feature representations in a Siamese network by predicting camera transformations from image pairs, [33] learn to group pixels that move together in a video. [50, 15] use a warping-based loss to learn depth from video. Recent work [36] leverages RGB-D based reconstruction [30] and is similar to this work, showing qualitatively impressive results learning a consistent low-dimensional labelling on a human dataset.

## 3   Method

This section discusses our method in detail, first introducing the general idea of dense equivariant labelling (section 3.1), and then presenting a concrete implementation of the latter using a novel deep CNN architecture (section 3.2).

### 3.1   Dense equivariant labelling

Consider a 3D object $S \subset \mathbb{R}^3$ or a class of such objects $S$ that are topologically isomorphic to a sphere $\mathcal{Z} \subset \mathbb{R}^3$ (i.e. the objects are simple closed surfaces without holes). We can construct a homeomorphism $p = \pi_S(q)$ mapping points of the sphere $q \in \mathcal{Z}$ to points $p \in S$ of the objects. Furthermore, if the objects belong to the same semantic category (*e.g.* faces), we can assume that these isomorphisms are *semantically consistent*, in the sense that $\pi_{S'} \circ \pi_S^{-1} : S \to S'$ maps points of object $S$ to semantically-analogous points in object $S'$ (*e.g.* for human faces the right eye in one face should be mapped to the right eye in another [39]).

While this construction is abstract, it shows that we can endow the object (or object category) with a spherical reference system $\mathcal{Z}$. The authors of [39] build on this construction to define a discrete system of object landmarks by considering a finite number of points $z_k \in \mathcal{Z}$. Here, we take the geometric embedding idea more literally and propose to explicitly learn a dense mapping from images of the object to the object-centric coordinate space $\mathcal{Z}$. Formally, we wish to learn a *labelling function* $\Phi : (\mathbf{x}, u) \mapsto z$ that takes a RGB image $\mathbf{x} : \Lambda \to \mathbb{R}^3, \Lambda \subset \mathbb{R}^3$ and a pixel $u \in \Lambda$ to the object point $z \in \mathcal{Z}$ which is imaged at $u$ (figure 1).

Similarly to [39], this mapping must be compatible or equivariant with image deformations. Namely, let $g : \Lambda \to \Lambda$ be a deformation of the image domain, either synthetic or due to a viewpoint change or other motion. Furthermore, let $g\mathbf{x} = \mathbf{x} \circ g^{-1}$ be the action of $g$ on the image (obtained by inverse warp). Barring occlusions and boundary conditions, pixel $u$ in image $\mathbf{x}$ must receive the same label as pixel $gu$ in image $g\mathbf{x}$, which results in the *invariance constraint*:

$$\forall \mathbf{x}, u : \quad \Phi(\mathbf{x}, u) = \Phi(g\mathbf{x}, gu). \tag{1}$$

Equivalently, we can view the network as a functional $\mathbf{x} \mapsto \Phi(\mathbf{x}, \cdot)$ that maps the image to a corresponding label map. Since the label map is an image too, $g$ acts on it by inverse warp.[1] Using this, the constraint (1) can be rewritten as the *equivariance relation* $g\Phi(\mathbf{x}, \cdot) = \Phi(g\mathbf{x}, \cdot)$. This can be visualized by noting that the label image deforms in the same way as the input image, as show for example in figure 3.

For learning, constraint (1) can be incorporated in a loss function as follows:

$$\mathcal{L}(\Phi|\alpha) = \frac{1}{|\Lambda|} \int_\Lambda \|\Phi(\mathbf{x}, u) - \Phi(g\mathbf{x}, gu)\|^2 \, du.$$

However, minimizing this loss has the significant drawback that a global optimum is obtained by simply setting $\Phi(\mathbf{x}, u) = $ const. The reason for this issue is that (1) is not quite enough to learn a useful object representation. In order to do so, we must require the labels not only to be equivariant, but also *distinctive*, in the sense that

$$\Phi(\mathbf{x}, u) = \Phi(g\mathbf{x}, v) \quad \Leftrightarrow \quad v = gu.$$

We can encode this requirement as a loss in different ways. For example, by using the fact that points $\Phi(\mathbf{x}, u)$ are on the unit sphere, we can use the loss:

$$\mathcal{L}'(\Phi|\mathbf{x}, g) = \frac{1}{|\Lambda|} \int_\Lambda \|gu - \mathrm{argmax}_v \langle \Phi(\mathbf{x}, u), \Phi(g\mathbf{x}, v) \rangle \|^2 \, du. \tag{2}$$

By doing so, the labels $\Phi(\mathbf{x}, u)$ must be able to discriminate between different object points, so that a constant labelling would receive a high penalty.

**Relationship with learning invariant visual descriptors.** As an alternative to loss (2), we could have used a pairwise loss[2] to encourage the similarity $\langle \Phi(\mathbf{x}, u), \Phi(\mathbf{x}', gu) \rangle$ of the labels assigned to corresponding pixels $u$ and $gu$ to be larger than the similarity $\langle \Phi(\mathbf{x}, u), \Phi(\mathbf{x}', v) \rangle$ of the labels assigned to pixels $u$ and $v$ that do *not* correspond. Formally, this would result in a pairwise loss similar to the ones often used to learn invariant visual descriptors for image matching. The reason why our method learns an object representation instead of a generic visual descriptor is that the *dimensionality* of the label space $\mathcal{Z}$ is just enough to represent a point on a surface. If we replace $\mathcal{Z}$ with a larger space such as $\mathbb{R}^d$, $d \gg 2$, we can expect $\Phi(\mathbf{x}, u)$ to learn to extract generic visual descriptors like SIFT instead. This establishes an interesting relationship between visual descriptors and object-specific coordinate vectors and suggests that it is possible to transition between the two by controlling their dimensionality.

## 3.2 Concrete learning formulation

In this section we introduce a concrete implementation of our method (figure 2). For the mapping $\Phi$, we use a CNN that receives as input an image tensor $\mathbf{x} \in \mathbb{R}^{H \times W \times C}$ and produces as output a label tensor $\mathbf{z} \in \mathbb{R}^{H \times W \times L}$. We use the notation $\Phi_u(\mathbf{x})$ to indicate the $L$-dimensional label vector extracted at pixel $u$ from the label image computed by the network.

The dimension of the label vectors is set to $L = 3$ (instead of $L = 2$) in order to allow the network to express uncertainty about the label assigned to a pixel. The network can do so by modulating the norm of $\Phi_u(\mathbf{x})$. In fact, correspondences are expressed probabilistically by computing the inner product of label vectors followed by the softmax operator. Formally, the probability that pixel $v$ in image $\mathbf{x}'$ corresponds to pixel $u$ in image $\mathbf{x}$ is expressed as:

$$p(v|u; \mathbf{x}, \mathbf{x}', \Phi) = \frac{e^{\langle \Phi_u(\mathbf{x}), \Phi_v(\mathbf{x}') \rangle}}{\sum_z e^{\langle \Phi_u(\mathbf{x}), \Phi_z(\mathbf{x}') \rangle}}. \tag{3}$$

In this manner, a shorter vector $\Phi_u$ results in a more diffuse probability distribution.

$$\mathcal{L}''(\Phi|\mathbf{x}, g) = \frac{1}{|\Lambda|} \int_\Lambda \max \left\{ 0, \max_v \Delta(u, v) + \langle \Phi(\mathbf{x}, u), \Phi(g\mathbf{x}, v) \rangle - \langle \Phi(\mathbf{x}, u), \Phi(g\mathbf{x}, gu) \rangle \right\} \, du,$$

where $\Delta(u, v) \geq 0$ is an error-dependent margin.

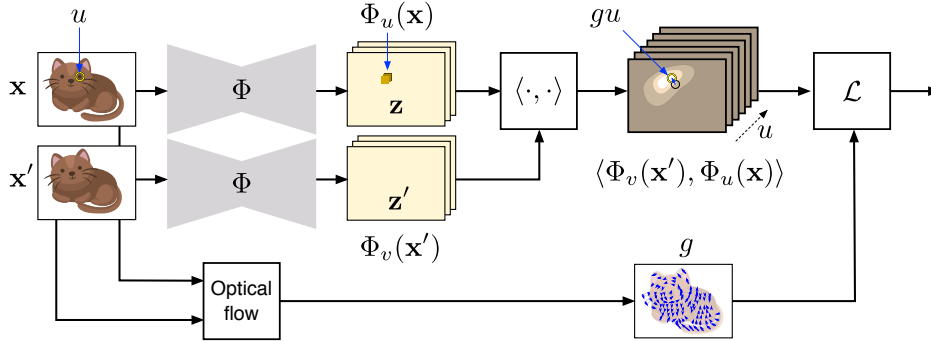

Figure 2: **Unsupervised dense correspondence network.** From left to right: The network $\Phi$ extracts label maps $\Phi_u(\mathbf{x})$ and $\Phi_v(\mathbf{x}')$ from the image pair $\mathbf{x}$ and $\mathbf{x}'$. An optical flow module (or ground truth for synthetic transformation) computes the warp (correspondence field) $g$ such that $\mathbf{x}' = g\mathbf{x}$. Then the label of each point $u$ in the first image is correlated to each point $v$ in the second, obtaining a number of score maps. The loss evaluates how well the score maps predict the warp $g$.

Next, we wish to define a loss function for learning $\Phi$ from data. To this end, we consider a triplet $\alpha = (\mathbf{x}, \mathbf{x}', g)$, where $\mathbf{x}' = g\mathbf{x}$ is an image that corresponds to $\mathbf{x}$ up to transformation $g$ (the nature of the data is discussed below). We then assess the performance of the network $\Phi$ on the triplet $\alpha$ using two losses. The first loss is the *negative log-likelihood* of the ground-truth correspondences:

$$\mathcal{L}_{\log}(\Phi|\mathbf{x}, \mathbf{x}', g) = -\frac{1}{HW} \sum_u \log p(gu|u; \mathbf{x}, \mathbf{x}', \Phi). \tag{4}$$

This loss has the advantage that it explicitly learns (3) as the probability of a match. However, it is not sensitive to the *size* of a correspondence error $v - gu$. In order to address this issue, we also consider the loss

$$\mathcal{L}_{\text{dist}}(\Phi|\mathbf{x}, \mathbf{x}', g) = \frac{1}{HW} \sum_u \sum_v \|v - gu\|_2^\gamma \, p(v|u; \mathbf{x}, \mathbf{x}', \Phi). \tag{5}$$

Here $\gamma > 0$ is an exponent used to control the robustness of the distance measure, which we set to $\gamma = 0.5, 1$.

**Nework details.** We test two architecture. The first one, denoted SIMPLE, is the same as [49, 39] and is a chain $(5, 20)_+, (2, \text{mp}), \downarrow_2, (5, 48)_+, (3, 64)_+, (3, 80)_+, (3, 256)_+, (1, 3)$ where $(h, c)$ is a bank of $c$ filters of size $h \times h$, $+$ denotes ReLU, $(h, \text{mp})$ is $h \times h$ max-pooling, $\downarrow_s$ is $s\times$ downsampling. Better performance can be obtained by increasing the support of the filters in the network; for this, we consider a second network DILATIONS $(5, 20)_+, (2, \text{mp}), \downarrow_2 , (5, 48)_+, (5, 64, 2)_+, (3, 80, 4)_+, (3, 256, 2)_+, (1, 3)$ where $(h, c, d)$ is a filter with $\times d$ dilation [43].

### 3.3 Learning from synthetic and true deformations

Losses (4) and (5) learn from triplets $\alpha = (\mathbf{x}, \mathbf{x}', g)$. Here $\mathbf{x}'$ can be either generated synthetically by applying a random transformation $g$ to a natural image $\mathbf{x}$ [39, 21], or it can be obtained by observing image pairs $(\mathbf{x}, \mathbf{x}')$ containing true object deformations arising from a viewpoint change or an object motion or deformation.

The use of synthetic transformations enables training even on static images and was considered in [39], who showed it to be sufficient to learn meaningful landmarks for a number of real-world object such as human and cat faces. Here, in addition to using synthetic deformations, we also consider using animated image pairs $\mathbf{x}$ and $\mathbf{x}'$. In principle, the learning formulation can be modified so that knowledge of $g$ is not required; instead, images and their warps can be compared and aligned directly based on the brightness constancy principle. In our toy video examples we obtain $g$ from the rendering engine, but it can in theory be obtained using an off-the-shelf optical flow algorithm which would produce a noisy version of $g$.

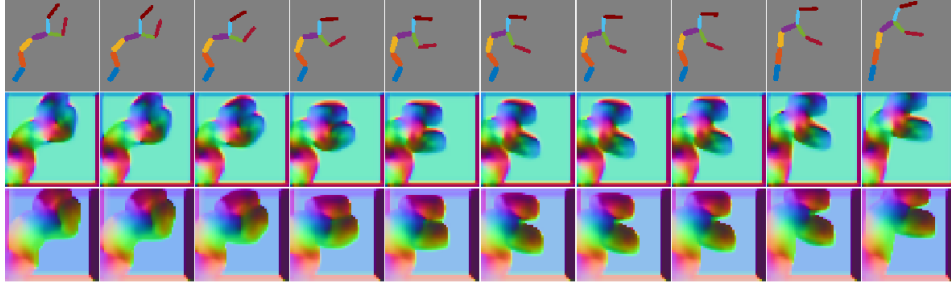

Figure 3: **Roboarm equivariant labelling.** Top: Original video frames of a simple articulated object. Middle and bottom: learned labels, which change equivariantly *with* the arm, learned using $\mathcal{L}_{log}$ and $\mathcal{L}_{dist}$, respectively. Different colors denote different points of the spherical object frame.

## 4    Experiments

This section assesses our unsupervised method for dense object labelling on two representative tasks: two toy problems (sections 4.1 and 4.2) and human and cat faces (section 4.3).

### 4.1    Roboarm example

In order to illustrate our method we consider a toy problem consisting of a simple articulated object, namely an animated robotic arm (figure 3) created using a 2D physics engine [38]. We do so for two reasons: to show that the approach is capable of labelling correctly deformable/articulated objects and to show that the spherical model $\mathcal{Z}$ is applicable also to thin objects, that have mainly a 1D structure.

**Dataset details.** The arm is anchored to the bottom left corner and is made up of colored capsules connected with joints having reasonable angle limits to prevent unrealistic contortion and self-occlusion. Motion is achieved by varying the gravity vector, sampling each element from a Gaussian with standard deviation $15\,\mathrm{m\,s}^{-2}$ every 100 iterations. Frames $\mathbf{x}$ of size $90 \times 90$ pixels and the corresponding flow fields $g : \mathbf{x} \mapsto \mathbf{x}'$ are saved every 20 iterations. We also save the positions of the capsule centers. The final dataset has 23999 frames.

**Learning.** Using the correspondences $\alpha = (\mathbf{x}, \mathbf{x}', g)$ provided by the flow fields, we use our method to learn an object centric coordinate frame $\mathcal{Z}$ and its corresponding labelling function $\Phi_u(\mathbf{x})$. We test learning $\Phi$ using the probabilistic loss (4) and distance-based loss (5). In the loss we ignore areas with zero flow, which automatically removes the background. We use the SIMPLE network architecture (section 3.2).

**Results.** Figure 3 provides some qualitative results, showing by means of colormaps the labels $\Phi_u(\mathbf{x})$ associated to different pixels of each input image. It is easy to see that the method attaches consistent labels to the different arm elements. The distance-based loss produces a more uniform embedding, as may be expected. The embeddings are further visualized in Figure 4 by projecting a number of video frames back to the learned coordinate spaces $\mathcal{Z}$. It can be noted that the space is invariant, in the sense that the resulting figure is approximately the same despite the fact that the object deforms significantly in image space. This is true for both embeddings, but the distance-based ones are geometrically more consistent.

**Predicting capsule centers.** We evaluate quantitatively the ability of our *object frames* to localise the capsule centers. If our assumption is correct and a coordinate system intrinsic to the object has been learned, then we should expect there to be a specific 3-vector in $\mathcal{Z}$ corresponding to each center, and our job is to find these vectors. Various strategies could be used, such as averaging the object-centric coordinates given to the centers over the training set, but we choose to incorporate the problem into the learning framework. This is done using the negative log-likelihood in much the same way as (4), limiting our vectors $u$ to the centers. This is done as an auxiliary layer with no backpropagation to the rest of the network, so that the embedding remains unsupervised. The error reported is the Euclidean distance as a percentage of the image width.

Results are given for the different loss functions used for unsupervised training in Table 1 and visualized in Figure 5 right, showing that the object centers can be located to a high degree of

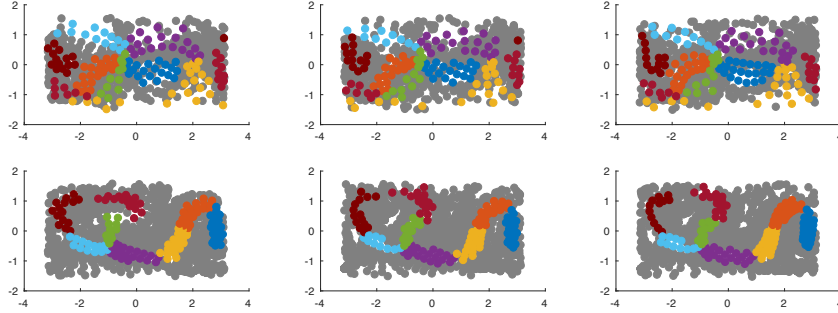

Figure 4: **Invariance of the object-centric coordinate space for Roboarm.** The plot projects frames 3,6,9 of figure 3 on the object-centric coordinate space $\mathcal{Z}$, using the embedding functions learned by means of the probabilistic (top) and distance (bottom) based losses. The sphere is then unfolded, plotting latitude and longitude (in radians) along the vertical and horizontal axes.

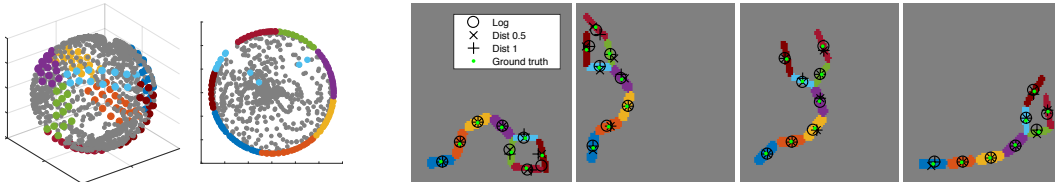

Figure 5: **Left: Embedding spaces of different dimension.** Spherical embedding (from the 3D embedding function $\Phi_u(\mathbf{x}) \in \mathbb{R}^3$) learned using the distance loss compared to a circular embedding with one dimension less. **Right: Capsule center prediction for different losses**.

accuracy. The negative log likelihood performs best while the two losses incorporating distance perform similarly.

We also perform experiments varying the dimensionality $L$ of the label space $\mathcal{Z}$ (Table 2). Perhaps most interestingly, given the almost one-dimensional nature of the arm, is the case of $L = 2$, which would correspond to an approximately circular space (since the length of vectors is used to code for uncertainty). As seen in the right of Figure 5 left, the segments are represented almost perfectly on the boundary of a circle, with the exception of the bifurcation which it is unable to accurately represent. This is manifested by the light blue segment trying, and failing, to be in two places at once.

| Unsupervised Loss | Error |
|---|---|
| $\mathcal{L}_{\log}$ | 0.97 % |
| $\mathcal{L}_{\text{dist}}, \gamma = 1$ | 1.13 % |
| $\mathcal{L}_{\text{dist}}, \gamma = 0.5$ | 1.14 % |

Table 1: Predicting capsule centers. Error as percent of image width.

| Descriptor Dimension | Error |
|---|---|
| 2 | 1.29 % |
| 3 | 1.14 % |
| 5 | 1.16 % |
| 20 | 1.28 % |

Table 2: Descriptor dimension ($\mathcal{L}_{\text{dist}}, \gamma = 0.5$). L>3 shows no improvement, suggesting L=3 is the natural manifold of the arm.

## 4.2 Textured sphere example

The experiment of Figure 6 tests the ability of the method to understand a complete rotation of a 3D object, a simple textured sphere. Despite the fact that the method is trained on pairs of adjacent video frames (and corresponding optical flow), it still learns a globally-consistent embedding. However, this required switching from from the SIMPLE to the DILATIONS architecture (section 3.2).

## 4.3 Faces

After testing our method on a toy problem, we move to a much harder task and apply our method to generate an object-centric reference frame $\mathcal{Z}$ for the *category* of human faces. In order to generate an image pair and corresponding flow field for training we warp each face synthetically using Thin Plate Spline warps in a manner similar to [39]. We train our models on the extensive CelebA [26] dataset of over 200k faces as in [39], excluding MAFL [49] test overlap from the given training split. It has annotations of the eyes, nose and mouth corners. Note that we do not use these to train our model. We also use AFLW [23], testing on 2995 faces [49, 42, 48] with 5 landmarks. Like [39] we

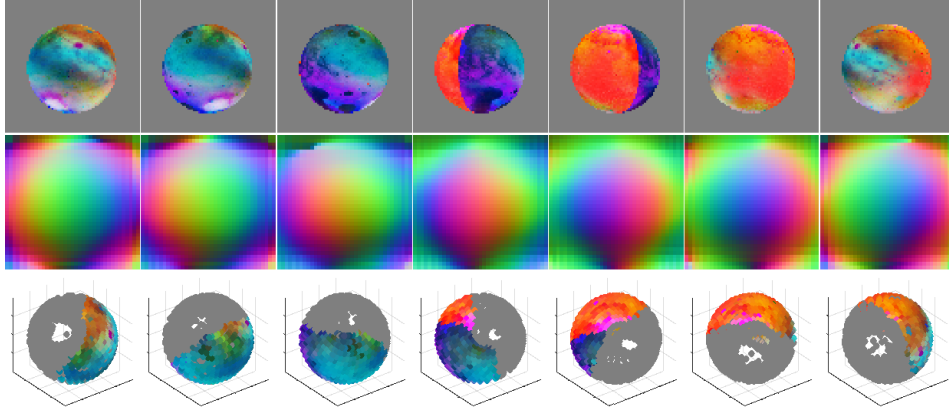

Figure 6: **Sphere equivariant labelling.** Top: video frames of a rotating textured sphere. Middle: learned dense labels, which change equivariantly *with* the sphere. Bottom: re-projection of the video frames on the object frame (also spherical). Except for occlusions, the reprojections are approximately invariant, correctly mapping the blue and orange sides to different regions of the label space

use 10,122 faces for training. We additionally evaluate qualitatively on a dataset of cat faces [47], using 8609 images for training.

**Qualitative assessment.** We find that for network SIMPLE the negative log-likelihood loss, while performing best for the simple example of the arm, performs poorly on faces. Specifically, this model fails to disambiguate the left and right eye, as shown in Figure 9 (right). The distance-based loss (5) produces a more coherent embedding, as seen in Figure 9 (left). Using DILATIONS this problem disappears, giving qualitatively smooth and unambiguous labels for both the distance loss (Figure 7) and the log-likelihood loss (Figure 8). For cats our method is able to learn a consistent object frame despite large variations in appearance (Figure 8).

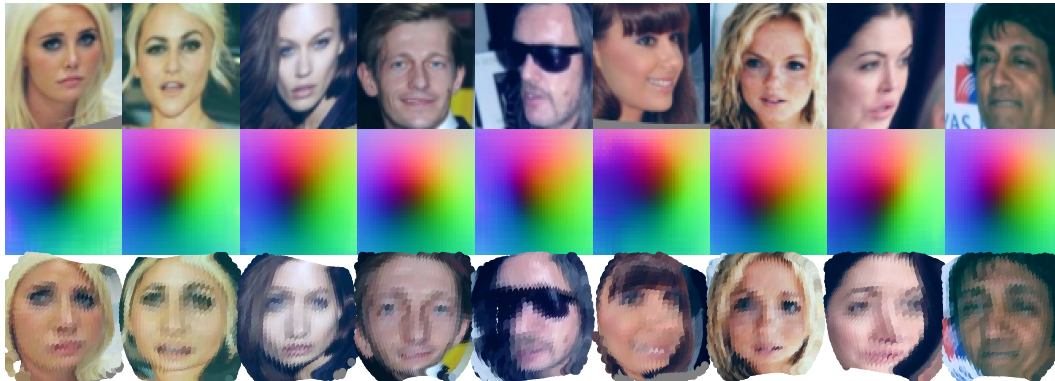

Figure 7: **Faces.** DILATIONS network with $\mathcal{L}_{\text{dist}}, \gamma = 0.5$. Top: Input images, Middle: Predicted dense labels mapped to colours, Bottom: Image pixels mapped to label sphere and flattened.

**Regressing semantic landmarks.** We would like to quantify the accuracy of our model in terms of ability to consistently locate manually annotated points, specifically the eyes, nose, and mouth corners given in the CelebA dataset. We use the standard test split for evaluation of the MAFL dataset [49], containing 1000 images. We also use the MAFL training subset of 19k images for learning to predict the ground truth landmarks, which gives a quantitative measure of the consistency of our *object frame* for detecting facial features. These are reported as Euclidean error normalized as a percentage of inter-ocular distance.

In order to map the object frame to the semantic landmarks, as in the case of the robot arm centers, we learn the vectors $z_k \in \mathcal{Z}$ corresponding to the position of each point in our canonical reference space and then, for any given image, find the nearest $z$ and its corresponding pixel location $u$. We report the localization performance of this model in Table 3 ("Error Nearest"). We empirically validate that with the SIMPLE network the negative log-likelihood is not ideal for this task (Figure 9) and

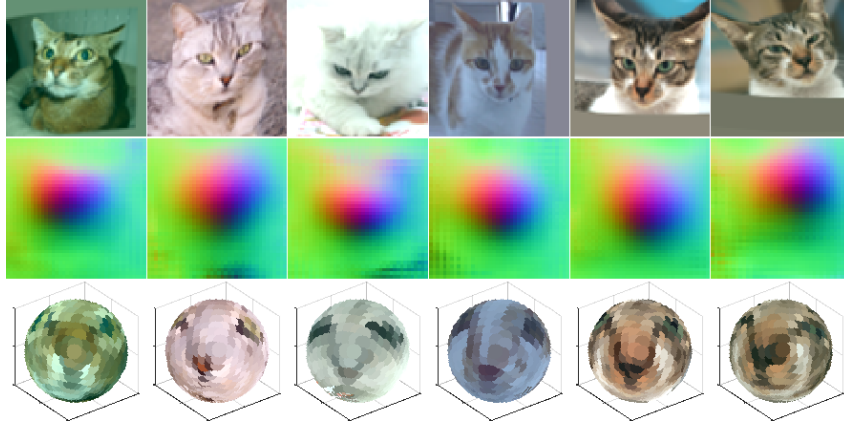

Figure 8: **Cats.** DILATIONS network with $\mathcal{L}_{\log}$. Top: Input images, Middle: Labels mapped to colours, Bottom: Images mapped to the spherical object frames.

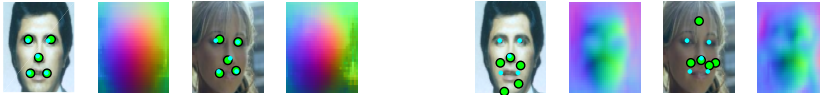

Figure 9: **Annotated landmark prediction** from the shown unsupervised label maps (SIMPLE network). **Left:** Trained with $\mathcal{L}_{dist}, \gamma = 0.5$, **Right:** Failure to disambiguate eyes with $\mathcal{L}_{log}$.
(Prediction: green, Ground truth: Blue)

we obtain higher performance for the robust distance with power 0.5. However, after switching to DILATIONS to increase the receptive field both methods perform comparably.

The method of [39] learns to regress $P$ ground truth coordinates based on $M > P$ unsupervised landmarks. By regressing from multiple points it is not limited to integer pixel coordinates. While we are not predicting landmarks as network output, we can emulate this method by allowing multiple points in our object coordinate space to be predictive for a single ground truth landmark. We learn one regressor per ground truth point, each formulated as a linear regressor $\mathbb{R}^{2M} \rightarrow \mathbb{R}^2$ on top of coordinates from $M = 50$ learned intermediate points. This allows the regression to say which points in $\mathcal{Z}$ are most useful for predicting each ground truth point.

We also report results after unsupervised finetuning of a CelebA network to the more challenging AFLW followed by regressor training on AFLW. As shown in Tables 3 and 4, we outperform other unsupervised methods on both datasets, and are comparable to fully supervised methods.

| Network | Unsup. Loss | Error Nearest | Error Regress |
|---|---|---|---|
| SIMPLE | $\mathcal{L}_{\log}$ | 75.02 % | — |
| SIMPLE | $\mathcal{L}_{dist}, \gamma = 1$ | 14.57 % | 7.94 % |
| SIMPLE | $\mathcal{L}_{dist}, \gamma = 0.5$ | 13.29 % | 7.18 % |
| DILATIONS | $\mathcal{L}_{\log}$ | 11.05 % | 5.83 % |
| DILATIONS | $\mathcal{L}_{dist}, \gamma = 0.5$ | 10.53 % | 5.87 % |
| [39] | | | 6.67 % |

Table 3: Nearest neighbour and regression landmark prediction on MAFL

| Method | Error |
|---|---|
| RCPR [5] | 11.6 % |
| Cascaded CNN [37] | 8.97 % |
| CFAN [45] | 10.94 % |
| TCDCN [49] | 7.65 % |
| RAR [42] | 7.23 % |
| Unsup. Landmarks [39] | 10.53 % |
| DILATIONS $\mathcal{L}_{dist}, \gamma = 0.5$ | 8.80 % |

Table 4: Comparison with supervised and unsupervised methods on AFLW

## 5  Conclusions

Building on the idea of viewpoint factorization, we have introduce a new method that can endow an object or object category with an invariant dense geometric embedding automatically, by simply observing a large dataset of unlabelled images. Our learning framework combines in a novel way the concept of equivariance with the one of distinctiveness. We have also proposed a concrete implementation using novel losses to learn a deep dense image labeller. We have shown empirically that the method can learn a consistent geometric embedding for a simple articulated synthetic robotic arm as well as for a 3D sphere model and real faces. The resulting embeddings are invariant to deformations and, importantly, to *intra-category* variations.

**Acknowledgments:** This work acknowledges the support of the AIMS CDT (EPSRC EP/L015897/1) and ERC 677195-IDIU. Clipart: FreePik.

## Footnotes

[1]In the sense that $g\Phi(\mathbf{x}, \cdot) = \Phi(\mathbf{x}, \cdot) \circ g^{-1}$.

[2]Formally, this is achieved by the loss

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
