[Reviews · NeurIPS 2017]

Reviewer 1



This paper extends the work in [30] to learn a dense object coordinate frame given pairs of images and dense correspondences between the pairs. The key insight is to include a distinctiveness constraint that encourages the correspondence error to be small when label vectors in the object coordinate frame are close. The approach is evaluated on a synthetic toy video sequence of a 2D robotic arm and a dataset of faces, and compared against [30] for sparse correspondences. Strengths: The approach is interesting and novel, as far as I’m aware. The writing and references look good. There aren’t major glaring weaknesses in my view. Potential weaknesses (if there’s space, please comment on the following in the rebuttal): W1. Loss (5) seems to have a degeneracy when v==gu, i.e., p(v|u) term is ignored. While this doesn’t seem to affect the results much, I’m wondering if the formulation could be slightly improved by adding a constant epsilon so that it’s (||v-gu||^\gamma + epsilon) * p(v|u). W2. L170-172 mentions that the learning formulation can be modified so that knowledge of g is not required. I find having g in the formulation somewhat unsatisfying, so I’m wondering if you can say a bit more about this possibility. W3. The shown examples are approximately planar scenes. It might be nice to include another toy example of a 3D object, e.g., textured sphere or other closed 3D object. Minor comments/suggested edits: + There are a number of small typos throughout; please pass through a spell+grammar checker. + Equation after L123 - It isn't clear what alpha is on first read, and is described later in the paper. Maybe introduce alpha here. + L259 mentions that the approach simply observes a large dataset of unlabelled images. However, flow is also required. Perhaps rephrase.

Reviewer 2



+ intersting idea - confusing exposition - insufficient experiments I like the idea of embedding all points in an image onto a sphere in order to do image alignment and keypoint tracking. I have not seen this before and the preliminary results seem to indicate that there is some promise. It's actually quite surprising that the idea works. All the authors do is say that the embedding should distort with the image. Why doesn't the network learn to embed different images as rotations of each other? Why does a consistent reference frame emerge? Would it work without optical flow examples, or does the optical flow do most of the work? The exposition of the paper is a bit confusing. Starting with the title: What is "object learning"? The authors might want to just call it embedding learning. What is a "dense equivariant image labeling"? Why is it "Unsupervised" learning when a "labeling" is involved? The authors might want to use the term weakly supervised. The language issues continue the the abstract. In the technical section some details are missing. How is (2) optimized? Or is it optimized at all? If not what is the connection to (4) and (5)? the main issue in the paper is the evaluation. The toy example is of little use, unless it's used to compare to prior work. The same is true for the main evaluation. There the authors compare to [30] (which works better), but they do not compare to other alignment baselines. How would the presented approach compare to [18] or [35] or vanilla optical flow? How sensitive is the presented approach to the training data? Does it heavily rely on optical flow, or would it work without it? Post rebuttal: The rebuttal clarified most of the concerns above. I like the new experiments, which should definitely find their way into the final paper.

Reviewer 3



Blue565 Unsupervised object learning from dense equivariant image labelling An impressive paper, marred by flaws in exposition, all fixable. The aim is to construct an object representation from multiple images, with dense labelling functions (from image to object), without supervision. Experiments seem to be very successful, though the paper would be improved by citing (somewhat) comparable numerical results on the MAFL dataset. The method is conceptually simple, which is a plus. The review of related methods seems good, though I admit to not knowing the field well enough to know what has been missed. Unfortunately, insufficient details of the method are given. In particular, there is no discussion of the relationships between the loss functions used in practice (in (4) and (5)) and those in the theoretical discussion (in page 4, first unnumbered equation and (2)). Also, I have some concerns about the authors’ use of the terms “equivariant” and “embedding” and their definition of “distinctive”. The standard definition of “equivariant” is: A function f is equivariant with respect to group of transformations if f(gx) = gf(x) for all x and all transformations g. (We note that the usage of “equivariant” in [30] is correct.) In contrast, f is invariant if f(gx) = f(x) for all x and all transformations g. Your definition in (1) says that \Phi is invariant with respect to image deformations. So please change the manuscript title! Note that both equivariance and invariance are a kind of “compatibility”. The word “embedding” usually means an embedding of an object into a larger space (with technical conditions). This doesn’t seem to be how the word is used here (though, OK, the sphere is embedded in R^3). The manuscript describes the map \Phi as an embedding. Perhaps “labelling” would be better, as in the title. The definition of “distinctive” on p4, line 124, is equivalent to: invariance of \Phi and injectivity of \Phi(x, \cdot) for every x. On p4, line 27: for a constant labelling, argmax is undefined, which is worrying. Overall the quality of the English is good, however there are several typos that should be fixed, a few of which are noted below. The paper is similar to [30], which is arXiv 1705.02193 (May 2017, unpublished as far as I know). They have significant overlap, however both the method and the datasets are different. —Minor comments. p1, line 17: typo: “form”->”from” p2, Figure 1 caption: - should be “Left: Given a large number of images” - typos: “coordiante”, “waprs” p4, first displayed equation: \alpha undefined. should be (x,g) p5, line 142: \Phi(x,u) has become \Phi_u x without comment. Fig 3 caption doesn’t explain clearly what the 2nd and 3rd rows are.